# Mechanism of Differential Susceptibility of Two (Canine Lung Adenocarcinoma) Cell Lines to 5-Aminolevulinic Acid-Mediated Photodynamic Therapy

**DOI:** 10.3390/cancers13164174

**Published:** 2021-08-19

**Authors:** Tomohiro Osaki, Narumi Kunisue, Urara Ota, Hideo Imazato, Takuya Ishii, Kiwamu Takahashi, Masahiro Ishizuka, Tohru Tanaka, Yoshiharu Okamoto

**Affiliations:** 1Joint Department of Veterinary Clinical Medicine, Faculty of Agriculture, Tottori University, Tottori 680-8553, Japan; yokamoto@tottori-u.ac.jp; 2SBI Pharmaceuticals Co., Ltd., Tokyo 106-6020, Japan; nkunisue@sbigroup.co.jp (N.K.); uota@sbigroup.co.jp (U.O.); himazato@sbigroup.co.jp (H.I.); taishii@sbigroup.co.jp (T.I.); kiwtakah@sbigroup.co.jp (K.T.); mishizuk@sbigroup.co.jp (M.I.); 3Neopharma Japan Co., Ltd., Tokyo 102-0071, Japan; o.tohru.tanaka@neopharmajp.com

**Keywords:** photodynamic therapy, lung adenocarcinoma, tumor, canine carcinoma

## Abstract

Photodynamic therapy (PDT) is a clinically approved, minimally invasive treatment for malignant tumors. Protoporphyrin IX (PpIX), derived from 5-aminolevulinic acid (5-ALA) as the prodrug, is one of the photosensitizers used in PDT. Recently, we reported a significant difference in response to 5-ALA-mediated PDT treatment in two canine primary lung adenocarcinoma cell lines (sensitive to PDT: HDC cells, resistant to PDT: LuBi cells). This study aimed to examine the difference in cytotoxicity of 5-ALA-mediated PDT in these cells. Although intracellular PpIX levels before irradiation were similar between HDC and LuBi cells, the percentage of ROS-positive cells and apoptotic cells in LuBi cells treated with 5-ALA-mediated PDT was significantly lower than that in HDC cells treated with 5-ALA-mediated PDT. A high dosage of the NO donor, DETA NONOate, significantly increased the cytotoxicity of 5-ALA-mediated PDT against LuBi cells. These results suggest that the sensitivity of 5-ALA-mediated PDT might be correlated with NO.

## 1. Introduction

Photodynamic therapy (PDT) has been described as a promising modality for treating malignant tumors or microorganisms [1,2,3,4]. PDT relies on the combination of a photosensitizer, which preferentially accumulates in malignant tissue, light of the appropriate wavelength, and oxygen. The interaction of visible and near-infrared light with the intracellular photosensitizer leads to reactive oxygen species (ROS) production [1,2,3,4]. Photosensitizers, such as porphyrins, chlorins, and phthalocyanines, have been studied for use in PDT [5,6,7,8]. One such photosensitizer is protoporphyrin IX (PpIX), derived from 5-aminolevulinic acid (5-ALA) as the prodrug.

PpIX induced by 5-ALA preferentially accumulates in the mitochondria of tumor cells [9]. It was suggested that one of the reasons for preferential PpIX accumulation was lower activity of ferrochelatase in tumor cells, compared to that in normal cells [10,11]. Ferrochelatase is responsible for the incorporation of ferrous iron into PpIX to form heme [12]. Therefore, a lower level of ferrochelatase cannot convert PpIX to heme, which results in the excessive accumulation of PpIX within tumor cells [10]. On the other hand, a higher level of ferrochelatase can convert PpIX to heme, which results in resistance to 5-ALA-mediated PDT. Moreover, the adenosine triphosphate (ATP)-binding cassette (ABC) transporter ABCG2, a porphyrin efflux pump, is usually downregulated in tumors. Thereby, it plays a significant role in regulating the cellular accumulation of PpIX in cancer cells [12]. It was reported that a higher level of ABCG2 expression tended to be associated with resistance to 5-ALA-mediated PDT. Indeed, the ABCG2 inhibitors, such as fumitremorgin C, improved the intracellular PpIX levels and the efficacy of 5-ALA-mediated PDT [13]. There have been some reports about the effectiveness of 5-ALA-mediated PDT in various cancer cell lines. These studies focused on the correlation between the intracellular accumulation of PpIX induced by exogenous 5-ALA and the effects of 5-ALA-mediated PDT in various human tumor cells [13,14,15].

Recently, we reported that the therapeutic effects of 5-ALA-mediated PDT were correlated with intracellular PpIX levels induced by 5-ALA in nine canine primary carcinoma cell lines [16]. Intracellular PpIX concentration did not significantly correlate with ABCG2 mRNA levels but showed a strong negative correlation with ferrochelatase mRNA levels [16]. Moreover, a profound discrepancy between the responses in the two canine primary lung adenocarcinoma cell lines (sensitive to PDT: HDC cells, resistant to PDT: LuBi cells) to 5-ALA-mediated PDT, in which intracellular PpIX levels were almost the same, was observed [16]. Recently, it was also reported that a higher level of glutathione peroxidase (GPx), which detoxifies lipid hydroperoxides, was correlated with resistance to PDT. MCF (human breast cancer cell line) cells, which have a higher level of GPx4, were more resistant to PDT than MDA-MB-231 (triple-negative human breast cancer cell line) cells, which have a lower level of GPx4 [17]. It was also reported that a low level of inducible nitric oxide synthase (iNOS)/nitric oxide (NO) could play a major role not only in resistance to PDT but also in enhanced aggressiveness of surviving tumor cells [18]. It is known that NO produced by iNOS/NOS2 in tumors stimulates proliferation, migration, and invasion of tumor cells and resistance to radiotherapy or chemotherapy [18].

To our knowledge, there are no reports on resistance to 5-ALA-mediated PDT in canine primary tumor cells. To clarify the general mechanisms of resistance to 5-ALA-PDT, we hypothesized that the efficacy of 5-ALA-mediated PDT would be associated with cellular GPx activity and NO levels in HDC and LuBi cells. In the present study, we aimed to investigate the cytotoxic activity of 5-ALA-mediated PDT combined with GPx4 inhibitor and NO donor in canine primary lung adenocarcinoma cell lines.

## 2. Materials and Methods

### 2.1. Chemicals

5-ALA hydrochloride was donated by SBI Pharmaceuticals Co., Ltd. (Tokyo, Japan). A stock solution of 100 mM 5-ALA in phosphate-buffered saline (PBS) (Nacalai Tesque Inc., Kyoto, Japan) was stored at −20 °C until it was used for the in vitro experiments.

### 2.2. Establishment of Primary Canine Lung Adenocarcinoma Cell Lines

We previously established two canine primary lung adenocarcinoma cell lines from the primary tumor tissues [16]. The two established canine primary lung adenocarcinoma cell lines, HDC and LuBi cells, were maintained in RPMI 1640 medium (Invitrogen, Carlsbad, CA, USA) supplemented with 10% heat-inactivated fetal bovine serum (FBS) (Nichirei Biosciences Inc., Tokyo, Japan) and penicillin–streptomycin–neomycin (PSN) solution (5 mg/mL penicillin, 5 mg/mL streptomycin, and 10 mg/mL neomycin; Invitrogen) in 5% CO_2_ at 37 °C. When cells reached about 80% confluence, primary cells were detached by treatment with 0.25% *w*/*v* trypsin-1 mmol/L EDTA·4Na solution with phenol red (FUJIFILM Wako Pure Chemical, Ltd., Osaka, Japan) at 37 °C. Trypsinization was stopped using RPMI 1640 containing 10% FBS. The cells were centrifuged and resuspended in RPMI 1640. Cell viability was assessed by the Trypan blue dye exclusion test. Then, the cells were seeded in 75 cm^2^ tissue culture flasks (Corning Inc., Durham, NC, USA).

### 2.3. Intracellular PpIX Concentration in Canine Primary Lung Adenocarcinoma Cell Lines

HDC and LuBi cells were seeded at 2–3 × 10^5^ cells into a 25 cm^2^ tissue culture flask (Corning Inc.) and incubated overnight. 5-ALA was added to the growth medium at a final concentration of 1 mM, and the HDC and LuBi cells were then incubated for 4 h. After changing the medium, the cells were irradiated with 630 nm LED light (Pleiades Technology LLC., Fukuoka, Japan) at an intensity of 20 mW/cm^2^ for 500 s (10 J/cm^2^). Immediately after irradiation, cells in the medium were collected. The nonirradiated and irradiated cells were washed twice with PBS and detached from the culture flask using trypsin. The cells in the medium and the detached cells were centrifuged at 300× *g* for 5 min at room temperature and were resuspended in PBS to a 1 × 10^6^/mL concentration. The cells were lysed using an ultrasonic homogenizer (Branson Sonifier 250, BRANSON Ultrasonics Co., Danbury, CT, USA). For the detection of PpIX, 0.2 mL of homogenate was shaken vigorously for 60 s with 0.02 mL of 50% *v*/*v* acetic acid and 0.9 mL of N,N-dimethylformamide-2-propanol solution (100:1 by vol.). The mixture was centrifuged at 13,150× *g* for 5 min at 4 °C to collect the supernatant. The supernatant liquids were analyzed with a high-performance liquid chromatography system using a Capcell Pak C18 UG120 column (Shiseido, Tokyo, Japan), mobile phase of acetonitrile-10 mM tetrabutylammonium hydroxide solution (pH 7.5) (70:30 by vol., flow rate, 1.0 mL/min; elution temperature, 40 °C), and fluorescence detector (excitation 400 nm, emission 630 nm) in the same way as in our previous reports [16].

### 2.4. Fluorescence Microscopic Imaging

HDC and LuBi cells were seeded at 1 × 10^5^ cells into 35 mm Petri dishes containing 2 mL of culture medium. After 24 h of incubation, the HDC and LuBi cells were incubated with 5-ALA (1 mM) for 4 h. After changing the medium, the cells were irradiated with an LED light source light (630 nm, 20 mW/cm^2^, 10 J/cm^2^). The nonirradiated and irradiated cells were imaged using fluorescence microscopy 10 min after irradiation. The fluorescence of PpIX was detected with a filter cube (excitation filter, 405/20 nm; emission filter, 640/30 nm) using a fluorescence microscope (BZ-X800, Keyence Co, Osaka, Japan).

### 2.5. Evaluation of the Cytotoxic Effects of Different 5-ALA Doses on Canine Primary Lung Adenocarcinoma Cell Lines

HDC and LuBi cells were seeded at 1–2 × 10^4^ cells per well into 96-well plates (Corning Inc.). Following 24 h of incubation, the HDC and LuBi cells were incubated with 5-ALA at various concentrations for 4 h. The final concentration of 5-ALA ranged from 0 to 1 mM. After changing the medium, the cells were irradiated with an LED light source light (630 nm, 20 mW/cm^2^, 10 J/cm^2^), similar to our previous reports [16]. The cells were then incubated for 24 h in the dark. According to the manufacturer’s instructions, cell survival following 5-ALA-mediated PDT was assessed using Cell Counting Kit-8 (Dojindo Molecular Technologies, Inc., Kumamoto, Japan).

### 2.6. Analysis of Apoptosis and Reactive Oxygen Species Induced by 5-ALA-Mediated PDT

HDC and LuBi cells were seeded at 1 × 10^4^ cells into 35 mm Petri dishes containing 2 mL culture medium. Following 24 h of incubation, the HDC and LuBi cells were incubated with 5-ALA (1 mM) for 4 h. After changing the medium, the cells were irradiated with an LED light source light (630 nm, 20 mW/cm^2^, 10 J/cm^2^). The cells were detached using trypsin 4 h after 5-ALA-mediated PDT. The relative percentage of ROS-positive and ROS-negative cells was measured using the Muse Oxidative Stress kit (Luminex Co., Austin, TX, USA). The percentage of apoptosis was measured using the Muse Caspase-3/7 Assay Kit and the Muse Annexin V and Dead Cell Assay Kit (Luminex Co.). Suspension cells were then loaded onto the Muse Cell Analyzer (Luminex Co.) according to the manufacturer’s instructions.

### 2.7. Determination of Cellular Glutathione Peroxidase Activity

HDC and LuBi cells were seeded at 1–2 × 10^5^ cells into 6-well plates (Corning Inc.). 5-ALA-mediated PDT was performed in the same manner as described in Section 2.6. The HDC and LuBi cells treated with/without PDT were washed twice with PBS and detached using trypsin. The HDC and LuBi cells were centrifuged, resuspended in PBS to a 2–3 × 10^5^/mL concentration, and then lysed using an ultrasonic homogenizer (Branson Sonifier 250). To assess cellular glutathione peroxidase activity, 50 μL of cell lysate was assessed spectrophotometrically using the Fluorimetric Glutathione Peroxidase Assay Kit (AAT Bioquest, Inc., Sunnyvale, CA, USA) according to the manufacturer’s protocol. In addition, the protein content in the cell lysate was measured spectrophotometrically using the TaKaRa BCA Protein Assay Kit (TaKaRa Bio Inc., Shiga, Japan) according to the manufacturer’s protocol.

### 2.8. Effect of Glutathione Peroxidase 4 Inhibitor on 5-ALA-Mediated PDT-Induced Cell Death

HDC and LuBi cells were seeded at 1–2 × 10^4^ cells per well of 96-well plates (Corning Inc.). Following 24 h of incubation, the HDC and LuBi cells were incubated with 5-ALA (1 mM) for 4 h in the presence (0, 1.25, 2.5, 5, and 10 μM) of the GPx4 inhibitor, RSL3 (MedChemExpress, Monmouth Junction, NJ, USA). After changing the medium, the cells were irradiated with an LED light source light (630 nm, 20 mW/cm^2^, 10 J/cm^2^). The cells were then incubated for 24 h in the dark. Cell survival following 5-ALA-mediated PDT was assessed using a Cell Counting Kit-8 according to the manufacturer’s instructions.

### 2.9. Analysis of Nitric Oxide-Positive Cells

HDC and LuBi cells were seeded at 1–2 × 10^5^ cells into each well of 6-well plates (Corning Inc.). 5-ALA-mediated PDT was performed in the same manner as described in Section 2.6. NO generation was assessed before and 4 h after laser irradiation using the Muse Nitric Oxide kit (Luminex Co.), according to the manufacturer’s protocols.

### 2.10. Effect of NO Donors on 5-ALA-Mediated PDT-Induced Cell Death

HDC and LuBi cells were seeded at 1–2 × 10^4^ per well of 96-well plates. Following 24 h of incubation, the HDC and LuBi cells were incubated with 5-ALA (1 mM) and various concentrations (0, 62.5, 125, and 250 mM) of the NO donors isosorbide dinitrate (ISDN) (Nitrol, Eisai Co. Ltd., Tokyo, Japan) and DETA NONOate (Cayman Chemical, Ann Arbor, MI, USA) for 4 h. After changing the medium, the cells were irradiated with an LED light source light (630 nm, 20 mW/cm^2^, 10 J/cm^2^). The cells were then incubated for 24 h in the dark. Cell survival following 5-ALA-mediated PDT was assessed using Cell Counting Kit-8 according to the manufacturer’s instructions.

### 2.11. Statistical Analysis

Data were analyzed using Sidak’s multiple comparison test and Mann–Whitney test. *p* < 0.05 was considered to indicate a statistically significant difference. Statistical analyses were performed using GraphPad Prism software (version 8.4.3; GraphPad Software, LLC., La Jolla, CA, USA).

## 3. Results

### 3.1. Intracellular PpIX Concentration in Canine Primary Lung Adenocarcinoma Cell Lines

In HDC cells, the intracellular PpIX concentration in cells treated with 5-ALA at 1 mM (PDT (−)) and those treated with a light dose of 10 J/cm^2^ and 1 mM 5-ALA (PDT (+)) was 23.0 ± 1.8 and 9.9 ± 0.8 nmol/g protein, respectively. In LuBi cells, the intracellular PpIX concentration of PDT (−) cells and PDT (+) cells was 22.6 ± 3.9 and 6.6 ± 0.8 nmol/g protein, respectively. In HDC (*p* = 0.0033) and LuBi (*p* = 0.0016) cells, the intracellular PpIX concentration of PDT (+) cells was significantly lower than that in PDT (−) cells (Figure 1).

### 3.2. Fluorescence Microscopic Imaging

In PDT (−) cells, most HDC and LuBi cells were strongly attached to the bottom of the dish (Figure 2A,E), and only some HDC and LuBi cells were rounded. In addition, PpIX fluorescence in HDC and LuBi PDT (−) cells appeared diffuse (Figure 2B,F).

In PDT (+) cells, approximately 50% of HDC cells showed a rounded morphology in the medium (Figure 2C), but many LuBi cells remained attached to the bottom of the dish (Figure 2G). PpIX fluorescence was photobleached in both primary tumor cell lines (Figure 2D,H).

### 3.3. Evaluation of the Cytotoxic Effects of Different 5-ALA Doses on Canine Primary Lung Adenocarcinoma Cell Lines

In HDC cells at 0.3 (*p* < 0.0001) and 1 (*p* < 0.0001) mM 5-ALA, the cell viabilities at a light dose of 10 J/cm^2^ were significantly lower than those at 0 J/cm^2^ (Figure 3A). In LuBi cells at 0.3 (*p* < 0.0001) and 1 (*p* < 0.0001) mM 5-ALA, the cell viabilities at a light dose of 10 J/cm^2^ were significantly higher than those at 0 J/cm^2^ (Figure 3B).

### 3.4. Analysis of Apoptosis and Reactive Oxygen Species Induced by 5-ALA-Mediated PDT

In HDC cells, the percentages of ROS-positive cells (*p* = 0.0286), caspase 3/7-positive cells (*p* = 0.0286), and annexin V-positive cells (*p* = 0.0286) in PDT (+) were significantly higher than those in PDT (−) (Figure 4A–C). In LuBi cells, the percentage of ROS-positive cells, caspase 3/7-positive cells, and annexin V-positive cells was not different between PDT (−) and PDT (+) (Figure 4A–C).

### 3.5. Determination of Cellular Glutathione Peroxidase Activity

In HDC cells, cellular GPx activity in the PDT (−) group was significantly higher than that in the PDT (+) group (*p* = 0.0178). In the PDT (+) groups, cellular GPx activity in LuBi cells was significantly higher than that in HDC cells (*p* = 0.0013) (Figure 5).

### 3.6. Effect of Glutathione Peroxidase 4 Inhibitor on 5-ALA-Mediated PDT-Induced Cell Death

The effect of a GPx4 inhibitor, RSL3, on 5-ALA-mediated PDT-induced cell death was investigated. In PDT (−) HDC cells, cell survival at 5 and 10 μM RSL3 was significantly lower than that at 0 μM RSL3 (*p* = 0.0003 and *p* < 0.0001, respectively). The same trend was observed at 1.25 μM RSL3 (*p* = 0.0003 and *p* < 0.0001, respectively) and 2.5 μM RSL3 (*p* = 0.0077 and *p* < 0.0001, respectively). Cell survival at 10 μM RSL3 was significantly lower than that at 5 μM RSL3 (*p* < 0.0001).

In PDT (+) HDC cells, cell survival at 1.25, 2.5, 5 and 10 μM RSL3 was significantly lower than that at 0 μM RSL3 (*p* < 0.0001, for all). Cell survival at 10 μM RSL3 was significantly lower than that at 2.5 and 5 μM RSL3 (*p* = 0.0002 and *p* = 0.0018, respectively) (Figure 6A).

In PDT (−) LuBi cells, cell survival at 1.25, 2.5, 5 and 10 μM RSL3 was significantly lower than that at 0 μM RSL3 (*p* < 0.0001, for all). In addition, cell survival at 2.5, 5 and 10 μM RSL3 was significantly lower than that at 1.25 μM RSL3 (*p* < 0.0001, for all), and cell survival at 10 μM RSL3 was significantly lower than at 2.5 μM RSL3 (*p* < 0.0001).

In PDT (+) LuBi cells, cell survival at 1.25, 2.5, 5 and 10 μM RSL3 was significantly lower than that at 0 μM RSL3 (*p* < 0.0001, for all). Cell survival at 2.5, 5 and 10 μM RSL3 was significantly lower than that at 1.25 μM RSL3 (*p* < 0.0001, for all) (Figure 6B).

### 3.7. Analysis of Nitric Oxide-Positive Cells

In HDC cells, the percentage of NO-positive cells in the PDT (+) group was significantly higher than that in the PDT (−) group (*p* = 0.0041). In the PDT (−) groups, the percentage of NO-positive cells in HDC cells was significantly higher than in LuBi cells (*p* = 0.0006). In the PDT (+) groups, the percentage of NO-positive cells in HDC cells was significantly higher than in LuBi cells (*p* < 0.0001) (Figure 7).

### 3.8. Effect of NO Donors on 5-ALA-Mediated PDT-Induced Cell Death

Different doses of ISDN did not affect cell viability in HDC and LuBi cells in PDT (−) and PDT (+). In HDC and LuBi cells in PDT (−), different doses of DETA NONOate did not affect cell viability. In HDC cells treated with PDT, DETA NONOate resulted in reduced cell viability in a dose-dependent manner (Figure 8A). The cell viabilities at 250 μM DETA NONOate were significantly lower than those in cells treated with 0, 62.5, and 125 μM DETA NONOate (*p* < 0.0001, *p* < 0.0001 and *p* =0.0149, respectively). The cell viabilities at 125 μM DETA NONOate were significantly lower than those in cells treated with 0 and 62.5 μM DETA NONOate (*p* = 0.0021 and *p* = 0.0397, respectively). In LuBi cells treated with PDT, the cell viabilities at 250 μM DETA NONOate were significantly lower than those in cells treated with 0, 62.5 and 125 μM DETA NONOate (*p* < 0.0001, for all) (Figure 8B).

## 4. Discussion

Photoactivation of PpIX in malignant cells initiates a photochemical reaction, generating high ROS levels, particularly singlet oxygen, which can induce cytotoxicity [10]. PpIX degrades to photoprotoporphyrin following irradiation due to photobleaching [19]. It was reported that PpIX fluorescence decayed by 63% due to photobleaching in well-differentiated human endometrial adenocarcinoma cells treated with 5-ALA and about 20–40 J/cm^2^ [20]. In the present study, intracellular PpIX levels before irradiation were similar between HDC (23.0 ± 1.8 nmol/g protein) and LuBi (22.6 ± 3.9 nmol/g protein) cells (Figure 1). After irradiation, intracellular PpIX levels in HDC (9.9 ± 0.8 nmol/g protein) and LuBi (6.6 ± 0.8 nmol/g protein) cell extracts were reduced to almost the same level as measured using HPLC (Figure 1). PpIX fluorescence levels in HDC and LuBi cells were decreased by 57.0% and 70.8%, respectively. It has been reported that PpIX is quickly photobleached [21]. Figure 2 also shows the photobleaching of PpIX post-PDT. Therefore, it was considered that photobleaching occurred in HDC and LuBi cells during irradiation.

Although a similar accumulation level of PpIX was observed in HDC and LuBi cells under the same conditions, there was a significant difference in photodamage observed in these cells. In HDC cells, PDT with 1 mM 5-ALA and subsequent irradiation with LED light (10 J/cm^2^) resulted in the generation of ROS, caspase-3/7 activation, and phosphatidylserine externalization. In contrast, the generation of ROS, caspase-3/7 activation, and phosphatidylserine externalization was not observed in LuBi cells. Although PpIX photobleaching occurred in LuBi cells, the percentage of ROS-positive cells did not increase. Therefore, it was considered that intracellularly generated ROS might be controlled by intracellular antioxidant molecules.

The most abundant intracellular antioxidant is the metabolic cofactor glutathione (GSH) [22]. Intracellular GSH is required to scavenge ROS with the aid of GPx, eliminating xenobiotics and lipid hydroperoxides [23]. GPx are enzymes crucial for detoxification and protecting cells from oxidative damage [24]. In the present study, LuBi cells were found to have a substantially higher level of GPx activity than HDC cells (Figure 5). It was considered that higher GPx activity in the LuBi cells might cause decreased sensitivity to 5-ALA-mediated PDT. In addition, the downregulation of GPx activity in HDC cells and unchanged GPx activity in LuBi cells after PDT was observed. It was considered that GPx activity was not affected by 5-ALA-mediated-PDT and might be related to cellular activity. To date, eight isoforms of the GPx family have been identified. GPx4 is found in both mitochondria and the cytosol [24]. It has been reported that GPx4 plays a role in maintaining the oxidative phosphorylation system and protecting mitochondria from oxidative damage [24,25].

Ferroptosis is induced mainly by the small molecules, erastin or RSL3 [26]. RSL3 inhibits GPx4 activity by covalent bonding and leads to lipid peroxide accumulation (Figure 9) [26,27]. It was reported that GPx4 inhibition resulted in ferroptotic death in therapy-resistant cancer cells [28]. The cytotoxicity of RSL3 in HDC cells was increased to over 5 μM RSL3 (Figure 6). In addition, the cytotoxicity of RSL3 in LuBi cells was increased to over 1.25 μM RSL3 (Figure 6). The difference in the cytotoxicity of RSL3 between the two primary tumor cell lines might be correlated with the difference in cellular GPx activity (Figure 5). It was reported that RSL3 enhanced cisplatin treatment for drug-resistant cells, compared with cisplatin treatment alone [29]. The mechanism was that cotreatment with RSL3 and cisplatin remarkably increased malondialdehyde and ROS levels, lipid oxidation, and sensitivity to cisplatin. In this study, although enhanced cytotoxicity due to the combined use of RSL3 and 5-ALA-mediated PDT was observed in HDC cells, this enhanced cytotoxicity was not observed in LuBi cells. Although GPx4 itself might play a role in the cytotoxicity in LuBi cells, it was considered that a mechanism other than that involving GPx4 might be associated with the resistance to 5-ALA-mediated PDT.

NO is a free radical molecule produced by three different NOS family enzymes: neuronal NOS/NOS1, iNOS/NOS2, and endothelial NOS/NOS3 [30,31]. Although the role of NO in tumor biology is still poorly understood, it was considered that NO was able to either enhance or diminish their biological effects, depending on NO concentration. It has been reported that low levels of endogenous NO promote tumor cell survival, persistence, and progression; they also mediate resistance to chemotherapy and radiotherapy [31,32]. Therefore, it was considered that the resistance of LuBi cells to 5-ALA-mediated PDT might be associated with low NO levels. However, high levels of NO result in cytotoxic effects, thereby promoting apoptosis of tumor cells [30,31,33]. It was considered that the sensitivity of HDC cells to 5-ALA-mediated PDT might be associated with high levels of NO. NO is not only involved in tumor progression but also has a major influence on the outcome of PDT [33,34]. It was reported that low levels of NO induced by a low PDT light dose were cytoprotective via the upregulation of the antiapoptotic proteins, NF-κB and Snail, but via the downregulation of Raf kinase inhibitor (RKIP). In contrast, high levels of NO induced by relatively high PDT light doses were cytotoxic via the downregulation of NF-κB and Snail but the upregulation of RKIP (Figure 9) [30,31,32]. Although NO levels were not evaluated in the present study, the percentage of NO-positive cells in HDC cells not treated with PDT was significantly higher than that in LuBi cells not treated with PDT (*p* = 0.0006). Furthermore, the percentage of NO-positive cells in HDC cells significantly increased after PDT (*p* = 0.0041), but the percentage of NO-positive cells in LuBi cells did not increase after PDT. It was suggested that the induction of NO by 5-ALA-mediated PDT in canine primary lung adenocarcinoma cell lines might play an essential role in its cytotoxicity considering these results. Therefore, an exogenous NO donor needs to be used to increase the cytotoxicity of 5-ALA-mediated PDT.

It was reported that the use of an NO donor significantly increased the cytotoxicity of pheophorbide a-mediated PDT [35]. In the present study, we investigated the cytotoxic activity of 5-ALA-mediated PDT combined with the NO donors ISDN and DETA NONOate. The cytotoxicity of ISDN with/without 5-ALA-mediated PDT in two primary tumor cell lines was not increased even at high doses of DETA NONOate. It was considered that ISDN is a low-potency nitrate [36,37]. On the other hand, the cytotoxicity of DETA NONOate on 5-ALA-mediated PDT in HDC cells sensitive to 5-ALA-mediated PDT was increased in a dose-dependent manner. In addition, the cytotoxicity of DETA NONOate on 5-ALA-mediated PDT to the resistance of LuBi cells to 5-ALA-mediated PDT was significantly increased at high doses of DETA NONOate (250 μM). It was considered that the difference might be correlated with the percentage of NO-positive cells at pre-PDT. The results showed that the efficacy of PDT was strengthened when 5-ALA was used in combination with an NO donor. However, we did not evaluate the level of NO but the percentage of NO-positive cells in this study. Therefore, evaluating the relationship between the effect of DETA NONOate and the level of NO should be considered in the future. Moreover, it might be difficult to reach the high concentration of 250 μM in in vivo settings. Therefore, using some kind of drug delivery system, such as liposomes, is necessary [38].

Finally, it was considered that, in contrast to single-agent treatment alone, the combination of 5-ALA-mediated PDT and an NO donor might result in significant modulation of the NF-κB/Snail/RKIP loop toward the expression of the inhibition of antiapoptotic NF-κB and Snail gene products and the upregulation of RKIP. In the future, we need to determine the expression levels of NF-κB, Snail, and RKIP by Western blotting to confirm this suggestion.

## 5. Conclusions

In the present study, we investigated the cytotoxicity of 5-ALA-mediated PDT in two primary canine lung adenocarcinoma cell lines. Intracellular PpIX levels before irradiation were similar between the two cell lines. Although photobleaching was observed in HDC and LuBi cells just after irradiation, cell survival after PDT differed between the two cell lines. In HDC cells, the cytotoxicity of 5-ALA-mediated PDT was dependent on 5-ALA concentration. However, under the experimental conditions, LuBi cells were resistant to 5-ALA-mediated PDT. Although GPx itself might be cytotoxic in LuBi cells, a mechanism other than GPx4 might be associated with resistance to 5-ALA-mediated PDT. The combined treatment using DETA NONOate and 5-ALA in HDC and LuBi cells in vitro showed that NO donors significantly increased the cytotoxicity of PDT in PDT-resistant tumor cells. However, future studies are warranted to analyze the cytotoxicity under various conditions and evaluate the relationship between the effect of DETA NONOate and the level of NO in in vitro studies. In addition, the in vivo efficacy of the combined treatment of an NO donor and 5-ALA-mediated PDT using some kind of drug delivery system should be studied.

## Figures and Tables

**Figure 1 cancers-13-04174-f001:**
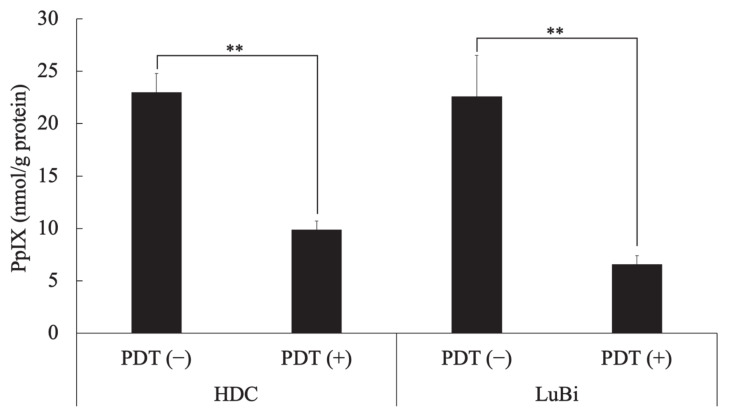
Intracellular PpIX concentration in canine primary lung adenocarcinoma cell lines. HDC and LuBi cells were seeded at 1 × 10^6^ into a 75 cm^2^ tissue culture flask and incubated overnight. The HDC and LuBi cells were then incubated with 1 mM 5-ALA for 4 h (PDT (−)). After changing the medium, the cells were irradiated with an LED light source light (630 nm, 20 mW/cm^2^, 10 J/cm^2^) (PDT (+)). Intracellular PpIX concentrations in the HDC and LuBi cells were analyzed with high-performance liquid chromatography. Results are presented as mean ± standard deviation (*n* = 3). Groups were compared using Sidak’s multiple comparison test, ** *p* < 0.01.

**Figure 2 cancers-13-04174-f002:**
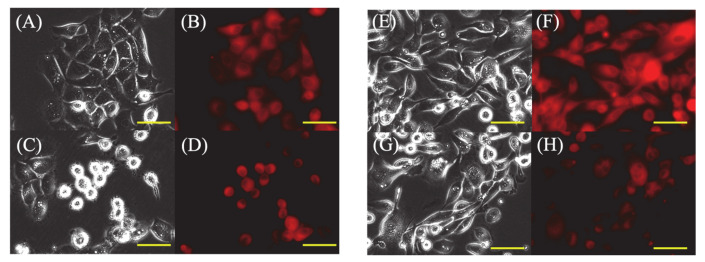
Fluorescence microscopic images of 5-ALA-induced protoporphyrin IX (PpIX) in HDC and LuBi cells. (**A**–**D**): HDC cell. (**E**–**H**): LuBi cells. (**A**,**C**,**E**,**G**): Bright field images. (**B**,**D**,**F**,**H**): PpIX fluorescence images. (**A**,**B**,**E**,**F**): PDT (−). (**C**,**D**,**G**,**H**): PDT (+). The HDC and LuBi cells were then incubated with 1 mM 5-ALA for 4 h (PDT (−)). After changing the medium, the cells are irradiated with an LED light source light (630 nm, 20 mW/cm^2^, 10 J/cm^2^) (PDT (+)). Cells were imaged using a fluorescence microscope 10 min after irradiation. In PDT (+) cells, approximately half of the HDC cells showed a rounded morphology in the medium, but many LuBi cells were still attached to the dish bottoms. PpIX fluorescence in each primary tumor cell type was photobleached. Bars = 50 μM.

**Figure 3 cancers-13-04174-f003:**
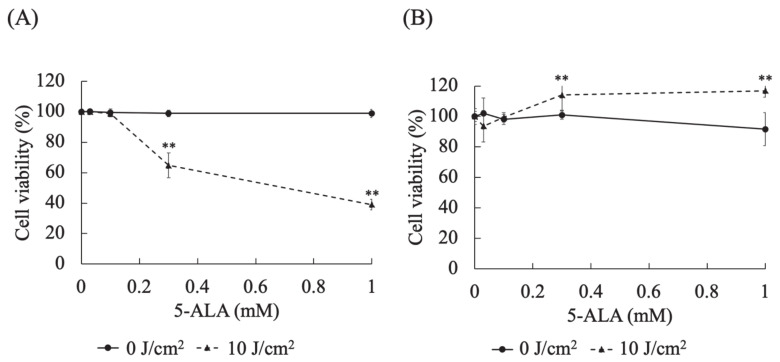
Evaluation of the cytotoxic effects of different 5-ALA doses on canine primary lung adenocarcinoma cell lines. (**A**): HDC cells. (**B**): LuBi cells. HDC and LuBi cells were seeded at 1–2 × 10^4^ cells per well into 96-well plates. Following 24 h of incubation, the HDC and LuBi cells were incubated with different concentrations of 5-ALA (0, 0.03, 0.1, 0.3, and 1 mM) for 4 h. After changing the medium, the cells were irradiated with an LED light source light (630 nm, 20 mW/cm^2^, 10 J/cm^2^). Results are presented as mean ± standard deviation (*n* = 6). Groups were compared using Sidak’s multiple comparison test, ** *p* < 0.01.

**Figure 4 cancers-13-04174-f004:**
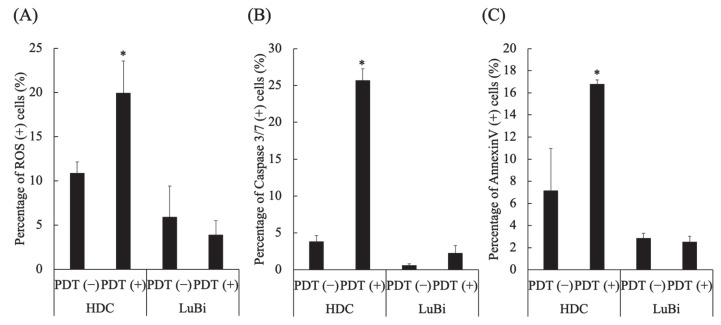
Analysis of apoptosis and reactive oxygen species (ROS) induced by 5-aminolevulinic acid-mediated photodynamic therapy (5-ALA-mediated PDT). (**A**) ROS induction by 5-ALA-mediated PDT. (**B**) Caspase 3/7 activation by 5-ALA-mediated PDT. (**C**) Apoptosis induction by 5-ALA-mediated PDT. Results are presented as mean ± standard deviation (*n* = 4). Groups were compared using the Mann–Whitney test, * *p* < 0.05 vs. HDC PDT (−).

**Figure 5 cancers-13-04174-f005:**
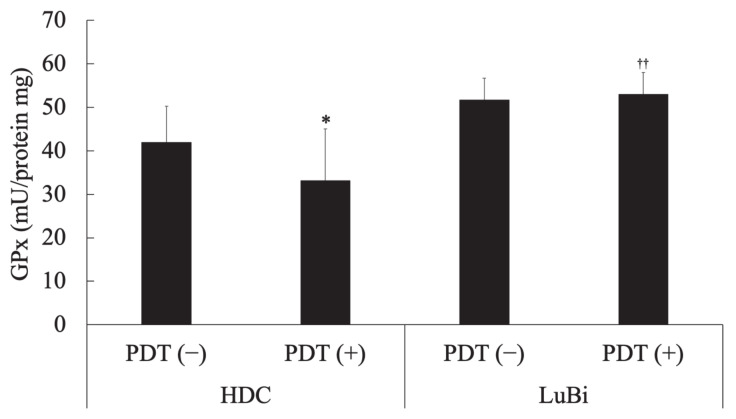
Determination of cellular glutathione peroxidase (GPx) activity. Cellular glutathione peroxidase activity was measured spectrophotometrically using the Fluorimetric Glutathione Peroxidase Assay Kit according to the manufacturer’s protocol. In addition, the protein content of the cell lysate was measured spectrophotometrically using the TaKaRa BCA Protein Assay Kit according to the manufacturer’s protocol. Results are presented as mean ± standard deviation (*n* = 6). Groups were compared using Sidak’s multiple comparison test, * *p* < 0.05 vs. HDC PDT (−) and ^††^ *p* < 0.01 vs. HDC PDT (+).

**Figure 6 cancers-13-04174-f006:**
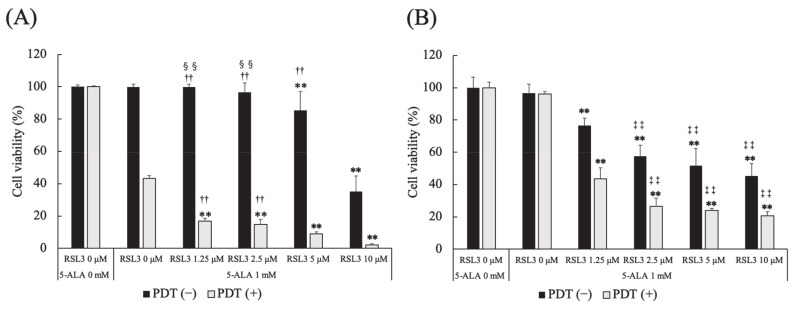
Effect of a glutathione peroxidase 4 (GPx4) inhibitor on 5-ALA-mediated PDT-induced cell death. (**A**) HDC and (**B**) LuBi cells. Results are presented as mean ± standard deviation (*n* = 6). Groups were compared using Sidak’s multiple comparison test, ** *p* < 0.01 vs. RSL3 0 μM. ^††^ *p* < 0.01 vs. RSL3 10 μM. ^‡‡^ *p* < 0.01 vs. RSL3 1.25 μM. ^§§^ *p* < 0.01 vs. RSL3 5 μM.

**Figure 7 cancers-13-04174-f007:**
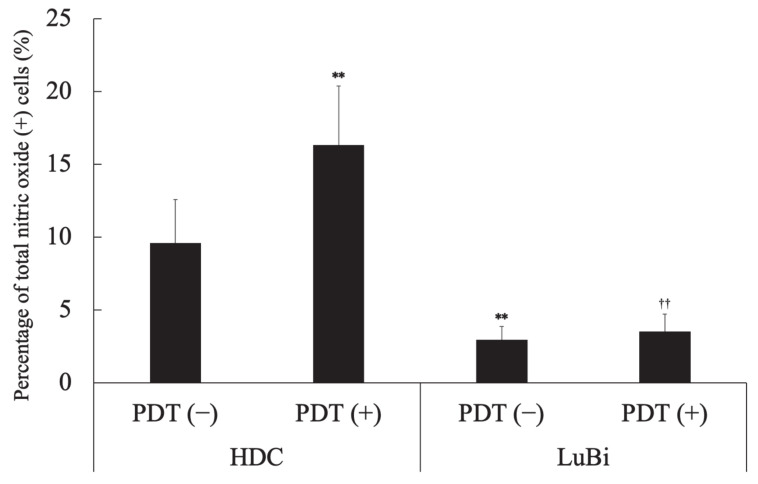
Analysis of nitric oxide (NO)-positive cells. NO generation was assessed before and 4 h after laser irradiation using the Muse Nitric Oxide kit, according to the manufacturer’s protocols. Results are presented as mean ± standard deviation (*n* = 6). Groups were compared using Sidak’s multiple comparison test, ** *p* < 0.01 vs. HDC PDT (−) and ^††^ *p* < 0.01 vs. HDC PDT (+).

**Figure 8 cancers-13-04174-f008:**
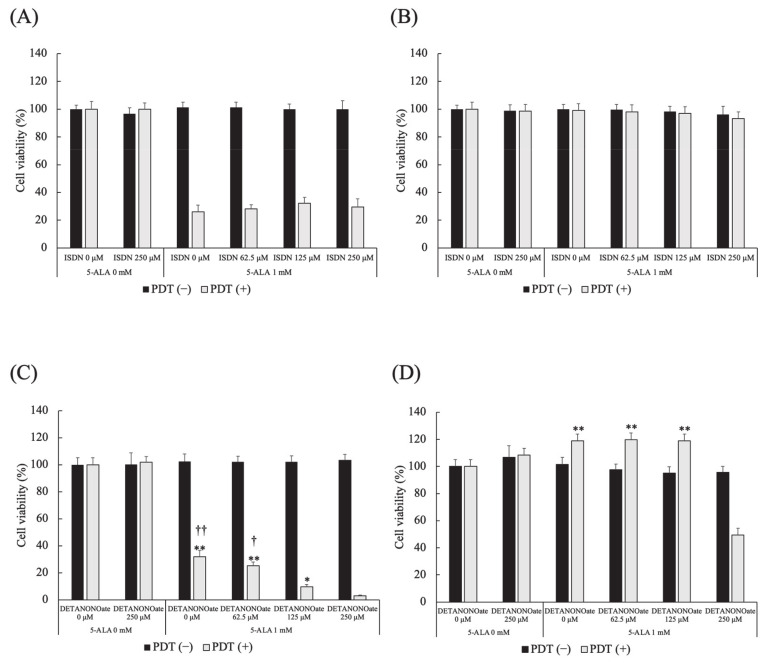
Effect of NO donors on 5-ALA-mediated PDT-induced cell death. (**A**,**B**) Isosorbide dinitrate (ISDN) and (**C**,**D**) DETA NONOate. (**A**,**C**) HDC and (**B**,**D**) LuBi cells. Results are presented as mean ± standard deviation (*n* = 6). Groups were compared using Sidak’s multiple comparison test, * *p* < 0.05, and ** *p* < 0.01 vs. DETA NONOate 250 μM. ^†^ *p* < 0.05, and ^††^ *p* < 0.01 vs. DETA NONOate 125 μM.

**Figure 9 cancers-13-04174-f009:**
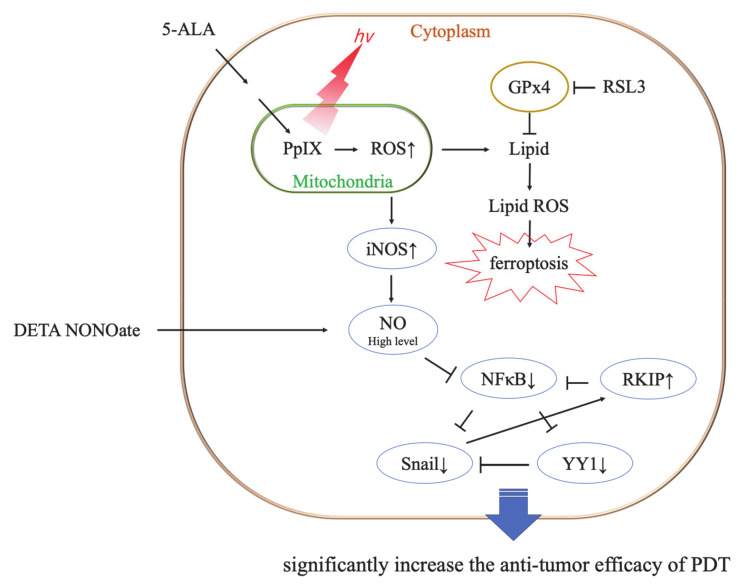
Representative mechanism of ferroptosis induced by photodynamic therapy (PDT), and the cytotoxic role of PDT and nitric oxygen (NO) donors. RSL3 inhibits the activity of glutathione peroxidase4 (GPx4) by covalent binding with GPx4 and induces ferroptosis [26]. No donor and PDT-induced high levels of NO result in significant antitumor cytotoxicity [32].

## Data Availability

The data presented in this study are available on request from the corresponding author.

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
