# Peer review of "Mechanism of Differential Susceptibility of Two (Canine Lung Adenocarcinoma) Cell Lines to 5-Aminolevulinic Acid-Mediated Photodynamic Therapy"

_cancers, 2021, doi:10.3390/cancers13164174_

Round 1

Reviewer 1 Report

Authors improved the manuscript in accordance with the key points/comments I made in the first round. 

Author Response

Comments and Suggestions for Authors

Authors improved the manuscript in accordance with the key points/comments I made in the first round. 

Response: Thank you for reviewing this manuscript and acknowledging improvements made.

Reviewer 2 Report

The authors have adressed most comments by adjusting tekst in the manuscript. The abstract and introduction are more logical and complete. Figure 9 helps understanding the proposed mechanisms, but maybe these mechanisms could already be explained in the results section to make it more understandable.

Regarding the results, I feel however that there are stilll too few experimental conditions to really propose a general mechanism.

  • The comment on subcellular localization based on the colocalization experiments with Rho123 is not addressed, as authors have removed this whole section instead of analyzing the results
  • The authors have chosen one light dose and state that this is based on preliminary results, however they also state that the clinical dose is much higher.Would it then be of value to test other light doses as well?

Author Response

Comments and Suggestions for Authors

The authors have adressed most comments by adjusting tekst in the manuscript. The abstract and introduction are more logical and complete. Figure 9 helps understanding the proposed mechanisms, but maybe these mechanisms could already be explained in the results section to make it more understandable.

Regarding the results, I feel however that there are stilll too few experimental conditions to really propose a general mechanism.

  • The comment on subcellular localization based on the colocalization experiments with Rho123 is not addressed, as authors have removed this whole section instead of analyzing the results

Response:

Thank you for your suggestion. We have re-written the sentence as follows, “PpIX induced by 5-ALA preferentially accumulates in mitochondria of tumor cells [9].”

  • The authors have chosen one light dose and state that this is based on preliminary results, however they also state that the clinical dose is much higher. Would it then be of value to test other light doses as well?

Response:

I think it is important to analyze cytotoxicity under various conditions for standardization of 5-ALA-PDT. However, in this study, we had wanted to compare the sensitivity for 5-ALA-PDT in HDC and LuBi cells. Therefore, we had analyzed the cytotoxicity under certain condition.

We have re-written the sentence as below:

“future studies are warranted to analyze the cytotoxicity under various conditions and evaluate the relationship between the effect of DETA NONOate and the level of NO in in vitro studies.”

Reviewer 3 Report

The authors have responded to all suggestions. There are a few remaining issues:

  1. In the abstract, mentioning the similar PpIX concentrations in the two cell lines is somehow lost. It is less important to note that both have shown photobleaching, the crucial point is that they had the same initial PpIX concentration – but responded so differently to irradiation.
  2. 4 lines 280-281: same p-values for all three comparisons?
  3. 6 lines 316-319: this seems to be wrong. You describe here the PDT (-) case, where this is not true and I also see no reasoning for providing two p-values.
  4. Please check lines 327-328 for correctness
  5. Figure 6 B is different from the original version in bar-heights. This can hardly be correct.
  6. Line 440f: this is not true, if I look to Figure 6B in whatever version!?

If these minor revisions are carefully considered, I recommend publication.

Author Response

Comments and Suggestions for Authors

The authors have responded to all suggestions. There are a few remaining issues:

  1. In the abstract, mentioning the similar PpIX concentrations in the two cell lines is somehow lost. It is less important to note that both have shown photobleaching, the crucial point is that they had the same initial PpIX concentration – but responded so differently to irradiation.

Response: Thank you for your recommendation. We have implemented the correction.

  1. 4 lines 280-281: same p-values for all three comparisons?

Response: Thank you for your suggestion. The p-values are the same for all three comparisons.

  1. 6 lines 316-319: this seems to be wrong. You describe here the PDT (-) case, where this is not true and I also see no reasoning for providing two p-values.

Response: We have effected the necessary corrections.

  1. Please check lines 327-328 for correctness

Response: Thank you for your suggestion. We have rechecked those. They are correct.

  1. Figure 6 B is different from the original version in bar-heights. This can hardly be correct.

Response: Thank you for your suggestion. As mentioned in the last submission, we did a re-experiment on LuBi cells and replaced the graph (Fig6 B).

  1. Line 440f: this is not true, if I look to Figure 6B in whatever version!?

Response:

Thank you for your suggestion. We have effected the necessary correction.

If these minor revisions are carefully considered, I recommend publication.

Round 2

Reviewer 2 Report

The authors have responded to my questions.

This manuscript is a resubmission of an earlier submission. The following is a list of the peer review reports and author responses from that submission.

Round 1

Reviewer 1 Report

The manuscript is devoted to the interesting and actual topic of the resistance of tumor cells to PDT and has basis on the previous successful research of the laboratory. However, the study looks logically incomplete. 

Major points:

  1. The abstract needs to be completely rewritten. It does not fit the focus of the article and is not self-contained. The statement of effects on Lines 18-21 suggests the photodynamic activity of 5-aminolevulinic acid, although the focus of the article is on the mechanisms of the differences in sensitivity between HDC and LuBi cells but not on describing the photosensitizing properties of 5-aminolevulinic acid. The NO-donor effect on sensitivity of LuBi cells stated at the end of the abstract is incomprehensible in isolation from the context and requires either an explanation after the conclusion, or introductory words at the beginning of the abstract. Lines 19-20: Abbreviations PDT (+) and PDT (-) need to be decoded before use, otherwise when reading only an abstract it is not immediately clear what is meant (light, sensitizer, light + sensitizer?).
  2. The meaning of Figure 2 and Section 3.2 is not clear for the logic of presentation in the article. Colocalization of Rho123 and PpIX is not visible at this low magnification. The figures also do not illustrate any difference between HDC and LuBi. The quantitative assessment of PplX in the studied cells is given in the previous figure.
  3. Has DETA NONOate worked as a NO donor enough in this model to raise the number of NO positive cells using chosen NO detection method? Was the donor effect of DETA NONOate at 125 mkM insufficient to cause effect? Obviously, it is required to control the level of NO (or at least the number of NO positive cells) during DETA NONOate treatement. The absence of a relationship between the effect and the level of NO such as the manifestation of sensitization with only one dose of the NO donor makes one strongly doubt the main conclusion. 
  4. Figure 6: Why RSL3 by itself (without 5-ALA-meditaed PDT) is significantly more toxic to LuBi than HDC? This point requires an explanation.

Minor points:

  1. Line 13: clically
  2. Line 327: "Therefore, it was considered that photobleaching was observed in HDC and LuBi cells just after irradiation". A very strange conclusion, as it seems obvious that photobleaching occurs right at the moment of exposure to light. 
  3. Lines 349-357: It is incorrect to use the term "anti-tumor effect". The data presented in Figure 6 were obtained on the cell culture and have nothing to do with the anti-tumor effect, which can only be assessed in vivo. 

Author Response

Reviewer 1

The manuscript is devoted to the interesting and actual topic of the resistance of tumor cells to PDT and has basis on the previous successful research of the laboratory. However, the study looks logically incomplete. 

Major points:

  1. The abstract needs to be completely rewritten. It does not fit the focus of the article and is not self-contained. The statement of effects on Lines 18-21 suggests the photodynamic activity of 5-aminolevulinic acid, although the focus of the article is on the mechanisms of the differences in sensitivity between HDC and LuBi cells but not on describing the photosensitizing properties of 5-aminolevulinic acid.

Response: Thank you for your suggestion. We have added the following sentence to the Abstract.

“Protoporphyrin IX, derived from 5-aminolevulinic acid (5-ALA) as the pro-drug, is one of the photosensitizers used in PDT.”

The NO-donor effect on sensitivity of LuBi cells stated at the end of the abstract is incomprehensible in isolation from the context and requires either an explanation after the conclusion, or introductory words at the beginning of the abstract.

Response: Thank you for your suggestion. We have re-written the text as shown below.

“Photodynamic therapy (PDT) is a clinically approved, minimally invasive treatment for malignant tumors. Protoporphyrin IX, derived from 5-aminolevulinic acid (5-ALA), as the pro-drug, is one of the photosensitizers used in PDT. Recently, we reported a significant difference in response to 5-ALA-mediated PDT treatment in two canine primary lung adenocarcinoma cell lines (sensitive to PDT: HDC cells, resistant to PDT: LuBi cells). This study aimed to examine the difference in cytotoxicity of 5-ALA-mediated PDT in these cells. Although photobleaching occurred in HDC and LuBi cells during irradiation, the percentage of ROS-positive cells and apoptotic cells in LuBi cells treated with 5-ALA-mediated PDT was significantly lower than that in HDC cells treated with 5-ALA-mediated PDT. Despite the increase in cytotoxicity due to the antioxidant enzyme inhibitor RSL3 in LuBi cells at concentrations of over 1.25 μM, enhanced cytotoxicity in the combined use of RSL3 and 5-ALA-mediated PDT was not observed in LuBi cells. A high dosage of the NO donor─DETA NONOate─which promotes apoptosis of tumor cells, significantly increased the cytotoxicity of 5-ALA-mediated PDT against LuBi cells. These results suggest that the sensitivity of 5-ALA-mediated PDT might be correlated with NO.”

Lines 19-20: Abbreviations PDT (+) and PDT (-) need to be decoded before use, otherwise when reading only an abstract it is not immediately clear what is meant (light, sensitizer, light + sensitizer?).

Response: Thank you for your suggestion. We have re-written it as shown below.

“PDT (+)” → “treated with 5-ALA-mediated PDT”

  1. The meaning of Figure 2 and Section 3.2 is not clear for the logic of presentation in the article. Colocalization of Rho123 and PpIX is not visible at this low magnification. The figures also do not illustrate any difference between HDC and LuBi. The quantitative assessment of PplX in the studied cells is given in the previous figure.

Response: Thank you for your suggestion. We have removed the experiment using Rho-123. We have qualitatively evaluated the photobleaching of PpIX, and therefore, revised the sentences in the results, as follows:

“In PDT (−) cells, most HDC and LuBi cells were strongly attached to the bottom of the dish (Fig. 2A,E), and only some HDC and LuBi cells were rounded. In addition, PpIX fluorescence in HDC and LuBi PDT (−) cells appeared diffuse (Fig. 2B,F).

 In PDT (+) cells, approximately 50% of HDC cells showed a rounded morphology in the medium (Fig. 2C), but many LuBi cells remained attached to the bottom of the dish (Fig. 2G). PpIX fluorescence was photobleached in both primary tumor cell lines (Fig. 2D,H).”

  1. Has DETA NONOate worked as a NO donor enough in this model to raise the number of NO positive cells using chosen NO detection method? Was the donor effect of DETA NONOate at 125 mkM insufficient to cause effect? Obviously, it is required to control the level of NO (or at least the number of NO positive cells) during DETA NONOate treatement. The absence of a relationship between the effect and the level of NO such as the manifestation of sensitization with only one dose of the NO donor makes one strongly doubt the main conclusion. 

Response: Thank you for your suggestion. We used isosorbide dinitrate as a NO donor in the clinical setting. However, isosorbide dinitrate did not enhance 5-ALA-mediated PDT for HDC and LuBi cells. Conversely, DETA NONOate could enhance 5-ALA-mediated PDT for LuBi cells, although at a higher dose. We think that DETA NONOate worked as a NO donor.

We have added the sentence, as follows:

“The cytotoxicity of ISDN with/without 5-ALA-mediated PDT in two primary tumor cell lines was not increased even at high doses of DETA NONOate. It was considered that ISDN is a low potency nitrate [35,36].”

and

“However, we did not evaluate the level of NO, but the percentage of NO positive cells in this study. Therefore, evaluating the relationship between the effect of DETA NONOate and the level of NO should be considered in the future.”

  1. Figure 6: Why RSL3 by itself (without 5-ALA-meditaed PDT) is significantly more toxic to LuBi than HDC? This point requires an explanation.

Response: Thank you for your suggestion. We have re-written the sentence, as shown below:

“The difference in the cytotoxicity of RSL3 between the two primary tumor cell lines might be correlated with the difference in cellular GPx activity (Fig. 5).”

Minor points:

  1. Line 13: clically

Response: Thank you for your suggestion. We have reworded it, as shown below:

“clically→“clinically”

  1. Line 327: "Therefore, it was considered that photobleaching was observed in HDC and LuBi cells just after irradiation". A very strange conclusion, as it seems obvious that photobleaching occurs right at the moment of exposure to light. 

Response: Thank you for your suggestion. We have re-written the sentence, as follows:

“Therefore, it was considered that photobleaching occurred in HDC and LuBi cells during irradiation.”

  1. Lines 349-357: It is incorrect to use the term "anti-tumor effect". The data presented in Figure 6 were obtained on the cell culture and have nothing to do with the anti-tumor effect, which can only be assessed in vivo. 

Response: Thank you for your suggestion. We have reworded it throughout the paper, as follows:

“anti-tumor effect”→“cytotoxicity”

Reviewer 2 Report

In the current manuscript, the authors investigate the sensitivity and resistance-mechanisms of two canine primary lung adenocarcinoma cell lines to 5-ALA mediated photodynamic therapy. The manuscript is well written and identification of resistance mechanisms to PDT are important to understand more about the photobiological aspects of this technique. However, in general I think the data described in this paper are limited and therefore not sufficient for publication in Cancers at this stage. It unfortunately lacks the size for creating an actual concept. Either the mechanisms should be investigated more thoroughly (activation of pathways, other inhibitors, other light dose (rates)), or the proposed mechanisms should be verified in other models (other cell lines, in vivo), as the authors address themselves as well in the last paragraph. Furthermore, some adjustments could improve the quality of the paper in my view:

General

  • Ferrochelatase and ABCG2 are two factors that play a role in sensitivity/resistance as described in the intro. They could be introduced more thoroughly, how does this work exactly? Furthermore the ferrochelatase status of the canine cell lines is discussed, however the ABCG2 status is not? Is this known for the used cell lines?
  • The mechanisms of resistance by NO/GPx are the main topic of this paper, but hardly introduced in the introduction. The explanation for the role of NO levels in cytotoxic and protective effects is not very clear. Addition of a Figure showing the NO/GPx4 pathway and how these could be involved in PDT resistance could help the reader understand the mechanisms.
  • Describe more clearly what the goal of this paper is. Discovery of general mechanisms of resistance that apply to the human situation as well?
  • Correct some typos, for example: described (line 30), established (line 75), mediated (line 265)

Results

  • Figure 2 : The authors mention that the PpIX and Rho123 derived fluorescence in adherent cells is quenched, but how do the authors know this, and compared to what the signal is quenched?
  • Figure 2 : In the HDC cells, it seems that more colocalization of the PpIX signal with the Rho-123 signal is observed, but authors do not describe or discuss this. Subcellular localization of the photosensitizer (membranous vs cytoplasm vs mitochondria etc) affects sensitivity a lot, so this aspect should be addressed in my view.
  • The authors use one light dose of 10 J/cm2 at 20 mw/cm2 in all experiments. What about trying an even higher light dose or dose rate? For example 50-100 J/cm2 at 100-200 mW/cm2, as is used in the clinics and other preclinical studies as well?
  • Figure 5 : If GPx activity is a mechanism for resistance, should it not be upregulated in the untreated LuBi cells when compared to the untreated HCD cells? And what is the mechanism for downregulation of GPx activity upon illumination in the HDC cells?
  • Figure 6 : Why does 1 mM 5-ALA combined with PDT induce cell death in this experiment (~70% viability) and not in the previous one (figure 3)? Is there a large variability between experiments?
  • Figure 6 : The authors state that RLS3 inhibition increases efficacy of PDT. For HDC cells, this this effect is present at 1,25-5 uM, however at 10 uM and for the Lubi cells at all concentrations, the addition of RLS alone without PDT also descreases cell viability. How can you distinguish the effect of RLS inhibition from that of the combination therapy?
  • Figure 8 : here, authors show that addiction of DETA NONOate increases the light induces toxicity of 5-ALA, in the HDC cells at 125 and 250 uM, and in LuBi cells only at the highest concentration of 250 uM. How relevant are these concentrations in vivo?

Author Response

Reviewer 2

In the current manuscript, the authors investigate the sensitivity and resistance-mechanisms of two canine primary lung adenocarcinoma cell lines to 5-ALA mediated photodynamic therapy. The manuscript is well written and identification of resistance mechanisms to PDT are important to understand more about the photobiological aspects of this technique. However, in general I think the data described in this paper are limited and therefore not sufficient for publication in Cancers at this stage. It unfortunately lacks the size for creating an actual concept. Either the mechanisms should be investigated more thoroughly (activation of pathways, other inhibitors, other light dose (rates)), or the proposed mechanisms should be verified in other models (other cell lines, in vivo), as the authors address themselves as well in the last paragraph. Furthermore, some adjustments could improve the quality of the paper in my view:

General 

  • Ferrochelatase and ABCG2 are two factors that play a role in sensitivity/resistance as described in the intro. They could be introduced more thoroughly, how does this work exactly? Furthermore the ferrochelatase status of the canine cell lines is discussed, however the ABCG2 status is not? Is this known for the used cell lines?

Response: Thank you for your comment. We have revised the Introduction as below.

“Ferrochelatase is responsible for the incorporation of ferrous iron into PpIX to form heme [11]. Therefore, a lower level of ferrochelatase cannot convert PpIX to heme, which results in the excessive accumulation of PpIX within tumor cells [9]. On the other hand, a higher level of ferrochelatase can convert PpIX to heme, which results in resistance to 5-ALA-mediated PDT. Moreover, the adenosine triphosphate (ATP)-binding cassette (ABC) transporter ABCG2, a porphyrin efflux pump, is usually downregulated in tumors. Thereby, it plays a significant role in regulating the cellular accumulation of PpIX in cancer cells [11].”

“Intracellular PpIX concentration did not significantly correlate with ABCG2 mRNA levels, but showed a strong negative correlation with ferrochelatase mRNA levels [15].”

  • The mechanisms of resistance by NO/GPx are the main topic of this paper, but hardly introduced in the introduction. The explanation for the role of NO levels in cytotoxic and protective effects is not very clear. Addition of a Figure showing the NO/GPx4 pathway and how these could be involved in PDT resistance could help the reader understand the mechanisms.

Response: Thank you for your comment. We have revised the introduction to reflect the title, as seen below:

“Recently, it was also reported that a higher level of glutathione peroxidase (GPx), which detoxifies lipid hydroperoxides, was correlated with resistance to PDT. MCF (human breast cancer cell line) cells, which have a higher level of GPx4, were more resistant to PDT than MDA-MB-231 (triple negative human breast cancer cell line) cells, which have a lower level of GPx4 [16]. It was also reported that a low level of inducible nitric oxide synthase (iNOS)/nitric oxide (NO) could play a major role not only in resistance to PDT but also in enhanced aggressiveness of surviving tumor cells [17]. It is known that NO produced by iNOS/NOS2 in tumors stimulates proliferation, migration, and invasion of tumor cells, and resistance to radiotherapy or chemotherapy [17].”

       We have added “Figure 9.”

“Representative mechanism of ferroptosis induced by photodynamic therapy (PDT), and the cytotoxic role of PDT and nitric oxygen (NO) donors. RSL3 inhibits the activity of glutathione peroxidase 4 (GPx4) by covalent binding with GPx4 and induces ferroptosis [25]. No donor and PDT-induced high levels of NO result in significant anti-tumor cytotoxicity [31].”

  • Describe more clearly what the goal of this paper is. Discovery of general mechanisms of resistance that apply to the human situation as well?

Response: Thank you for your suggestion. We have re-written the text, as below:

“To clarify the general mechanisms of resistance to 5-ALA-PDT, we hypothesized that …”

  • Correct some typos, for example: described (line 30), established (line 75), mediated (line 265)

Response: Thank you for your suggestion. We have corrected the typos, as shown below:

“describved"→“described”

“establsied "→“established”

“meditaed”→“mediated”

Results

  • Figure 2 : The authors mention that the PpIX and Rho123 derived fluorescence in adherent cells is quenched, but how do the authors know this, and compared to what the signal is quenched?

Thank you for your suggestion. We have compared the intensity of fluorescence obtained from fluorescence imaging. Therefore, we have reworded it as below.

“quenched”→“photobleached”

  • Figure 2 : In the HDC cells, it seems that more colocalization of the PpIX signal with the Rho-123 signal is observed, but authors do not describe or discuss this. Subcellular localization of the photosensitizer (membranous vs cytoplasm vs mitochondria etc) affects sensitivity a lot, so this aspect should be addressed in my view.

Response: Thank you for your suggestion. We have removed the experiment using Rho-123. We have qualitatively evaluated the photobleaching of PpIX.

Therefore, we have revised the sentences in the results, as seen below:

“In PDT (−) cells, most HDC and LuBi cells were strongly attached to the bottom of the dish (Fig. 2A,E), and only some HDC and LuBi cells were rounded. In addition, PpIX fluorescence in HDC and LuBi PDT (−) cells appeared diffuse (Fig. 2B,F).

 In PDT (+) cells, approximately 50% of HDC cells showed a rounded morphology in the medium (Fig. 2C), but many LuBi cells remained attached to the bottom of the dish (Fig. 2G). PpIX fluorescence was photobleached in both primary tumor cell lines (Fig. 2D,H).”

  • The authors use one light dose of 10 J/cm2 at 20 mw/cm2 in all experiments. What about trying an even higher light dose or dose rate? For example 50-100 J/cm2 at 100-200 mW/cm2, as is used in the clinics and other preclinical studies as well?

Response: Thank you for your suggestion. Based on preliminary tests results, we used one light dose of 10 J/cm2 at 20 mW/cm2. In the in vitro study, the experimental conditions were sufficient to induce cell death and enable the identification of the differences in cell death. In the in vitro and clinical settings, we usually use a light dose of 100~150 J/cm2 at 200 mW/cm2.  

  • Figure 5 : If GPx activity is a mechanism for resistance, should it not be upregulated in the untreated LuBi cells when compared to the untreated HCD cells? And what is the mechanism for downregulation of GPx activity upon illumination in the HDC cells?

Response: Thank you for your question. Although we consider that GPx activity is not the main mechanism for resistance, it is likely that GPx activity in LuBi cells is higher than that in HDC cells. We hypothesized that higher GPx activity has a role in the resistance of cells to PDT because GPx enzymes are crucial for the detoxification and protection of cells from oxidative damage.

The downregulation of GPx activity in HDC cells upon illumination might be related to the decrease in the cellular activity of HDC cells after PDT.

  • Figure 6 : Why does 1 mM 5-ALA combined with PDT induce cell death in this experiment (~70% viability) and not in the previous one (figure 3)? Is there a large variability between experiments?

Response: Thank you for your suggestion. Because there may have been technical differences, we repeated the experiment on LuBi cells. Further, we have changed to reproducible data (Fig. 6B).

  • Figure 6 : The authors state that RLS3 inhibition increases efficacy of PDT. For HDC cells, this this effect is present at 1,25-5 uM, however at 10 uM and for the Lubi cells at all concentrations, the addition of RLS alone without PDT also descreases cell viability. How can you distinguish the effect of RLS inhibition from that of the combination therapy?

Response: Thank you for your suggestion. As described in the Discussion, the cytotoxicity of RSL3 in HDC cells was increased at over 5 μM. In addition, the cytotoxicity of RSL3 in LuBi cells was increased at over 1.25 μM (Fig. 6). Enhanced cytotoxicity of the combined use of RSL3 and 5-ALA-mediated PDT was not observed in LuBi cells. We considered that a mechanism other than that involving GPx4 might be associated with the resistance to 5-ALA-mediated PDT.

  • Figure 8 : here, authors show that addiction of DETA NONOate increases the light induces toxicity of 5-ALA, in the HDC cells at 125 and 250 uM, and in LuBi cells only at the highest concentration of 250 uM. How relevant are these concentrations in vivo?

Response: Thank you for your suggestion. I think it might be difficult to reach the high concentration of 250 μM in in vivosettings. Therefore, we have added the sentences, as shown below:

“Moreover, it might be difficult to reach the high concentration of 250 μM in in vivo settings. Therefore, using some kind of drug delivery system, such as liposomes, is necessary [37].”

Reviewer 3 Report

The authors have previously observed that two cell lines responded very differently to ALA-PDT, although their capacity to accumulate intracellular PpIX was very comparable. In the present manuscript they report on investigations about the possibly underlying causes. Obviously there are considerable differences in the production of ROS and induction of apoptosis and necrosis. They also studied GPx activity and the effect of its inhibition, finally conclude that the difference in PDT response is best explained by a lack of nitric oxide induction in the less responsive cells. 
I am not sure, whether this is indeed true or the only responsible mechanism, as the data show a single bar only (Fig 8B), which indicates the supportive action of an NO donor in the less responsive cells. This is too little to draw a final conclusion. Nevertheless, the observation and investigations provide some insight in a rarely discussed phenomenon and should be made available to the PDT community after some revision.
The most severe problem I have with the investigations is the different conditions, under which the experiments were made: for the determination of PpIX and GPx, the cells were washed with PBS immediately after PDT, whereas for the survival assays, cells were continuously incubated. For ROS etc. determination, the procedure is unclear (detached using trypsin, but previously washed?). The problem is that quite a few sensitized and irradiated cells are detached from the flask bottom immediately after irradiation. These will be removed from further investigation by washing, whereas some of them might reattach, if further incubated before being further processed some time later. This might deteriorate any meaningful comparison or conclusion. 
Use of English need thorough workup please. Not being a native speaker myself, I make only a few comments on this.

Figure captions: I suggest to delete unnecessary (the more trivial) M&M parts (e.g. cell numbers seeded, size of flasks, “after changing the medium”. The journal wants “short explanatory title and caption”.
Title: I suggest to write “Mechanism of differential susceptibility of two (canine lung adenocarcinoma) cell lines..” instead of “Mechanism of sensitivity of canine lung adenocarcinoma cells…”
L13  clinically
L15 cells > cell lines (there are several instances, where this replacement should be done, please check)
L18 to 21 you miss pointing out the difference between the two cell lines!
L19 need to explain, what PDT(+) and PDT(-) means
L23 what is an anti-tumor effect? There is no tumor in cell culture experiments. I think, it most often just “cytotoxicity” what you mean (many instances, please check!)
L24 this sentence just repeats the previous one!?
L25 “however”?
L30 described
L34 to 36 sentence makes little sense
L40 reads as if ferrochelatase were the only reason for tumor selectivity, may add “mainly” due to…or “one of the reasons”…
L43 and 46 might look for references outside your group
L45 which inhibitor?
L56 resistance > resistant
L57 representative of many instances: when you write “it was considered…” you sometimes mean the hypothesis or conclusion you draw from the measurements described in this manuscript, sometimes your own previous results and sometimes results in referenced literature. This is difficult to follow. I want to encourage you to make clear statements. 
L58 this cell line (singular here)
L59 if I understand correctly, the upregulation of GPx, but lower levels of NO increase resistance? Your sentence reads as if resistance increases with both parameters increasing
L72 I am not sure whether this remains stable for more than very few weeks.
L75 to 77 …established two primary canine… from tumor tissues [15]. HDC and …
L90 may delete “the cells”
L91 was the ALA incubation in absence or presence of FBS?
L92 supplier of LED source? May delete “intensity of” or add “an”
L104 and l125 in the same way as in our
L115 delete “an”
L135 “for experiments”?
L190 you mixed PDT(-) and PDT(+). Please add “cells”. 
Fig. 1: I have a problem with the PDT(+) bars: as you wash away the detached cells, their height may no longer mean a lot. It is difficult to conclude that LuBi photobleached as much as HDC did. May be true, maybe not.
L200 your images do NOT show exclusively “strongly attached” PDT(-) LuBi cells! 
L202 to 206 this description is very bad, sorry. 1. You do not mention the important differences between the two cell lines 2. “on the one hand”, but there is no “other”. 3. I would not say PpIX fluorescence is quenched (many instances), but photobleached. 4. The sentence starting at l 205 does not refer to anything.
L235 and l236 may combine into one sentence, as all three signals go in the same direction. 
Fig 5 isnt it also (even more) important that HDC and LuBi differ in GPx in the PDT(-) groups? And that, in contrast to HDC, there is no difference for LuBi in GPx between PDT(-) and PDT(+)?
L257 mediated (several instances)
L262 two stars at p < 0.01!
L265 may consider to introduce the name of the inhibitor by writing “…of GPx4 inhibitor RSL3 on …”
L265 you mean “investigated” rather than “estimated”?
L267 and l271 may want to not repeat p < 0.0001 four times?
L265 to 272 you miss describing the effects of RSL3 on LuBi PDT(-)! The significance indicators in Fig 6B are rather meaningless in view of the (dark) cytotoxicity of RSL3 on 5-ALA incubated cells. You try to describe that in the discussion. The description part of this should be part of the results, I suggest. It needs thorough rewording, however (see there).
L298 “treated without PDT” may find a better expression…
L313 to 316 I do not completely agree. Apart from that, tumor cell selectivity of PpIX accumulation is not the crucial point of your investigations – on the contrary!
L327 to 328 this sentence does not make much sense to me
L348 to 357 this paragraph needs to be completely rewritten. Do have an idea, why RSL3 is so much more toxic to LuBi cells than to HDC cells?
L373 “estimated”?
L358ff I do not properly understand the NO mediated pathways. I just cannot resolve the contradiction of your observation that an NO donor should improve PDT, whereas others claim the contrary and improve PDT effect by blocking NO production (Fahey and Girotti, Cancers 11/2, 2019). May please explain to me – and in the manuscript if necessary.
L398 may first state that PpIX levels were similar
L399 between the two cell lines
L401 this is a very week statement
L405 it is not true that your data show that the NO donor improved PDT efficacy only in the PDT resistant cell line. On the contrary! It worked better in the HDCs.
L415 ??

Author Response

Reviewer 3

The authors have previously observed that two cell lines responded very differently to ALA-PDT, although their capacity to accumulate intracellular PpIX was very comparable. In the present manuscript they report on investigations about the possibly underlying causes. Obviously there are considerable differences in the production of ROS and induction of apoptosis and necrosis. They also studied GPx activity and the effect of its inhibition, finally conclude that the difference in PDT response is best explained by a lack of nitric oxide induction in the less responsive cells. 
I am not sure, whether this is indeed true or the only responsible mechanism, as the data show a single bar only (Fig 8B), which indicates the supportive action of an NO donor in the less responsive cells. This is too little to draw a final conclusion. Nevertheless, the observation and investigations provide some insight in a rarely discussed phenomenon and should be made available to the PDT community after some revision.

Response: Thank you for your suggestion.

We used isosorbide dinitrate as an NO donor in the clinical setting. However, isosorbide dinitrate did not enhance 5-ALA-mediated PDT for HDC and LuBi cells. Conversely, DETA NONOate could enhance 5-ALA-mediated PDT for LuBi cells, although at a higher dose. We consider that DETA NONOate worked as an NO donor.

The most severe problem I have with the investigations is the different conditions, under which the experiments were made: for the determination of PpIX and GPx, the cells were washed with PBS immediately after PDT, whereas for the survival assays, cells were continuously incubated. For ROS etc. determination, the procedure is unclear (detached using trypsin, but previously washed?).

Response: Thank you for your comment.

・Regarding timing of measurements

We assessed intracellular PpIX concentration, GPx activity, and NO positive cells before and 4 h after laser irradiation to estimate short-term changes. Conversely, cytotoxicity induced by 5-ALA-mediated PDT was assessed 24 hours after PDT, because more time is required to assess cell growth inhibition.

・We have revised the sentence below to clarify regarding the washed cells.

“Immediately after irradiation, cells in the medium were collected. The non-irradiated and irradiated cells were washed twice with PBS and detached from the culture flask using trypsin. The cells in the medium and the detached cells were centrifuged at 300 × g for 5 min at room temperature and then resuspended in PBS to a 1 × 106/mL concentration.”

The problem is that quite a few sensitized and irradiated cells are detached from the flask bottom immediately after irradiation. These will be removed from further investigation by washing, whereas some of them might reattach, if further incubated before being further processed some time later. This might deteriorate any meaningful comparison or conclusion. 
Response: Thank you for your comment.

Each examination was conducted separately. Because we did not further incubate tumor cells, they did not re-attach to the bottom of the flask.

Use of English need thorough workup please. Not being a native speaker myself, I make only a few comments on this.

Figure captions: I suggest to delete unnecessary (the more trivial) M&M parts (e.g. cell numbers seeded, size of flasks, “after changing the medium”. The journal wants “short explanatory title and caption”.

Response: Thank you for your suggestion. We have removed unnecessary M&M parts.

Title: I suggest to write “Mechanism of differential susceptibility of two (canine lung adenocarcinoma) cell lines..” instead of “Mechanism of sensitivity of canine lung adenocarcinoma cells…”

Response: Thank you for your suggestion. We have revised the title, as follows:

“Mechanism of differential susceptibility of two (canine lung adenocarcinoma) cell lines to 5-aminolevulinic acid-mediated photodynamic therapy.”

L13  clinically

Response: Thank you for the suggestion. We have reworded it, as suggested.

“clically→“clinically”

L15 cells > cell lines (there are several instances, where this replacement should be done, please check)

Response: Thank you for the suggestion. We have reworded it throughout the manuscript, as seen below:

“lung adenocarcinoma cells”→“lung adenocarcinoma cell lines”

L18 to 21 you miss pointing out the difference between the two cell lines!

Response: Thank you for the comment. We have re-written the sentence, as follows:

“This study aimed to examine the difference in cytotoxicity of 5-ALA-mediated PDT in these cells. Although photobleaching occurred in HDC and LuBi cells during irradiation, the percentage of ROS-positive cells and apoptotic cells in LuBi cells treated with 5-ALA-mediated PDT was significantly lower than that in HDC cells treated with 5-ALA-mediated PDT.”

L19 need to explain, what PDT(+) and PDT(-) means

Response: Thank you for your suggestion. We have re-written the sentence as below.

e.g. Section 3.1.

“In HDC cells, the intracellular PpIX concentrations in cells treated with 5-ALA at 1 mM {PDT (−)} and those with a light dose of 10 J/cm2and 1 mM 5-ALA {PDT (+)}...”

L23 what is an anti-tumor effect? There is no tumor in cell culture experiments. I think, it most often just “cytotoxicity” what you mean (many instances, please check!)

Response: Thank you for your suggestion. We have reworded it throughout the paper as below.

“anti-tumor effect”→“cytotoxicity”

L24 this sentence just repeats the previous one!?

Response: Thank you for your suggestion. We have re-written the sentence as below.

“These results suggest that the sensitivity of 5-ALA-mediated PDT might be correlated with NO.”

L25 “however”?

Response: Thank you for the suggestion. We have removed “However.”

L30 described

Response: Thank you for your suggestion. We have reworded it as below.

“describved"→“described”

L34 to 36 sentence makes little sense

Response: Thank you for the suggestion. We have re-written the sentence, as follows:

“Photosensitizers, such as porphyrins, chlorins, and phthalocyanines, have been studied for use in PDT [5–8].”

L40 reads as if ferrochelatase were the only reason for tumor selectivity, may add “mainly” due to…or “one of the reasons”…

Response: Thank you for the suggestion. We have re-written the sentence, as seen below:

“It was suggested that one of the reasons for preferential PpIX accumulation was lower activity of ferrochelatase in tumor cells, compared to that in normal cells [9,10].”

L43 and 46 might look for references outside your group

Response: Thank you for the suggestion. We have changed these references and cited those outside our group.

L45 which inhibitor?

Response: Thank you for the suggestion. We have added, “such as fumitremorgin C.”

L56 resistance > resistant

Response: Thank you for the suggestion. We have reworded it, as follows:

“resistance”→“resistant”

L57 representative of many instances: when you write “it was considered…” you sometimes mean the hypothesis or conclusion you draw from the measurements described in this manuscript, sometimes your own previous results and sometimes results in referenced literature. This is difficult to follow. I want to encourage you to make clear statements. 

Response: Thank you for the suggestion. We have removed “It was considered that some resistance mechanisms might exist for 5-ALA-mediated PDT in these cell lines.”

L58 this cell line (singular here)

Response: Thank you for the suggestion. We have removed “It was considered that some resistance mechanisms might exist for 5-ALA-mediated PDT in these cell lines.”

L59 if I understand correctly, the upregulation of GPx, but lower levels of NO increase resistance? Your sentence reads as if resistance increases with both parameters increasing

Response: Thank you for the suggestion. We have revised the sentence, as seen below:

“…a low level of inducible nitric oxide synthase (iNOS)/nitric oxide (NO) could play a major role, not only in resistance to PDT, but also in enhanced aggressiveness of surviving tumor cells [17].”

L72 I am not sure whether this remains stable for more than very few weeks.

Response: Thank you for the comment. We made a mistake in the description of the temperature. We have revised the sentence, as follows:

“…was stored at -20°C, until it was used for the in vitro experiments.”

L75 to 77 …established two primary canine… from tumor tissues [15]. HDC and …

Response: Thank you for your suggestion. We have revised the sentence, as follows:

“We have previously established two canine primary lung adenocarcinoma cell lines from the primary tumor tissues [15]. The two established canine primary lung adenocarcinoma cell lines, HDC and LuBi cells…”

L90 may delete “the cells”

Response: Thank you for the suggestion. We have deleted “the cells.”

L91 was the ALA incubation in absence or presence of FBS?

Response: Thank you for the suggestion. We have re-written the sentence, as seen below:

“5-ALA was added to the growth medium at a final concentration of 1 mM, and the HDC and LuBi cells were then incubated for 4 h.”

L92 supplier of LED source? May delete “intensity of” or add “an”

Response: Thank you for your comment. We have rewritten the sentence, as follows:

“…the cells were irradiated with 630 nm LED light (Pleiades Technology LLC., Fukuoka, Japan) at an intensity of 20 mW/cm2 for 500 sec (10 J/cm2).”

L104 and l125 in the same way as in our

Response: Thank you for your suggestion. We have rewritten the sentence, as follows:

“in a same way”→“in the same way as in our”

L115 delete “an”

Response: Thank you for your comment. We have removed “an.”

L135 “for experiments”?

Response: Thank you for your suggestion. We have removed “for experiments.”

L190 you mixed PDT(-) and PDT(+). Please add “cells”. 

Response: Thank you for your suggestion. We have added “the cells.”

Fig. 1: I have a problem with the PDT(+) bars: as you wash away the detached cells, their height may no longer mean a lot. It is difficult to conclude that LuBi photobleached as much as HDC did. May be true, maybe not.

Response: Thank you for your suggestion. We have rewritten the sentence as follows.

“Immediately after irradiation, cells in the medium were collected. The non-irradiated and irradiated cells were washed twice with PBS and detached from the culture flask using trypsin. The cells in the medium and the detached cells were centrifuged at 300 × g for 5 min at room temperature and were resuspended in PBS to a 1 × 106/mL concentration.”

L200 your images do NOT show exclusively “strongly attached” PDT(-) LuBi cells! 

Response: Thank you for your suggestion. We have rewritten the sentence, as follows:

“In PDT (−) cells, most HDC and LuBi cells were strongly attached to the bottom of the dish, and only some HDC and LuBi cells were rounded.”

L202 to 206 this description is very bad, sorry. 1. You do not mention the important differences between the two cell lines 2. “on the one hand”, but there is no “other”. 3. I would not say PpIX fluorescence is quenched (many instances), but photobleached. 4. The sentence starting at l 205 does not refer to anything.

Response: Thank you for your suggestion.

1 and 4. We have rewritten the sentences, as follows:

“In PDT (−) cells, most HDC and LuBi cells were strongly attached to the bottom of the dish (Fig. 2A,E), and only some HDC and LuBi cells were rounded. In addition, PpIX fluorescence in HDC and LuBi PDT (−) cells appeared diffuse (Fig. 2B,F).

 In PDT (+) cells, approximately 50% of HDC cells showed a rounded morphology in the medium (Fig. 2C), but many LuBi cells remained attached to the bottom of the dish (Fig. 2G). PpIX fluorescence was photobleached in both primary tumor cell lines (Fig. 2D,H)..”

  1. We have removed the sentence.
  2. We have reworded it, as follows:

“quenched”→“photobleached”

L235 and l236 may combine into one sentence, as all three signals go in the same direction. 

Response: Thank you for the suggestion. We have merged and rewritten the sentence, as follows:

“In HDC cells, the percentages of ROS-positive cells (p = 0.0286), caspase 3/7-positive cells (p = 0.0286), and annexin V-positive cells (p = 0.0286) in PDT (+) were significantly higher than those in PDT (−) (Fig. 4 A–C).”

Fig 5 isnt it also (even more) important that HDC and LuBi differ in GPx in the PDT(-) groups? And that, in contrast to HDC, there is no difference for LuBi in GPx between PDT(-) and PDT(+)?
L257 mediated (several instances)

Response: Thank you for your suggestion. We have corrected all instances of “meditaed” to “mediated”.

L262 two stars at p < 0.01!

Response: Thank you for your suggestion. We have revised the expression, as follows:

“*p < 0.05 and **p < 0.01.”

L265 may consider to introduce the name of the inhibitor by writing “…of GPx4 inhibitor RSL3 on …”

Response: Thank you for your suggestion. We have revised the sentence, as follows:

“The effect of a GPx4 inhibitor, RSL3, on 5-ALA-mediated PDT-induced cell death was investigated.”

L265 you mean “investigated” rather than “estimated”?

Response: Thank you for your suggestion. We have removed “estimated.”

L267 and l271 may want to not repeat p < 0.0001 four times?

Response: Thank you for your suggestion. We have rewritten the sentence, as follows:

“(p < 0.0001, for all)”

L265 to 272 you miss describing the effects of RSL3 on LuBi PDT(-)! The significance indicators in Fig 6B are rather meaningless in view of the (dark) cytotoxicity of RSL3 on 5-ALA incubated cells. You try to describe that in the discussion. The description part of this should be part of the results, I suggest. It needs thorough rewording, however (see there).

We have rewritten the sentences, as follows:

“The effect of a GPx4 inhibitor, RSL3, on 5-ALA-mediated PDT-induced cell death was investigated. In PDT (−) HDC cells, cell survival at 5 and 10 μM RSL3 was significantly lower than that at 0 μM RSL3 (p = 0.0003 and p < 0.0001, respectively). The same trend was observed at 1.25 μM RSL3 (p = 0.0003 and p < 0.0001, respectively) and 2.5 μM RSL3 (p = 0.0077 and p < 0.0001, respectively). Cell survival at 10 μM RSL3 was significantly lower than at 5 μM RSL3 (p < 0.0001).

In PDT (+) HDC cells, cell survival at 1.25, 2.5, 5, and 10 μM RSL3 was significantly lower than that at 0 μM RSL3 (p < 0.0001, for all). Cell survival at 10 μM RSL3 was significantly lower than that at 2.5 and 5 μM RSL3 (p = 0.0002 and p = 0.0018, respectively) (Fig. 6A).

In PDT (−) LuBi cells, cell survival at 1.25, 2.5, 5, and 10 μM RSL3 was significantly lower than that at 0 μM RSL3 (p < 0.0001, for all). In addition, cell survival at 2.5, 5, and 10 μM RSL3 was significantly lower than at 1.25 μM RSL3 (p < 0.0001, for all), and cell survival at 10 μM RSL3 was significantly lower than that at 2.5 μM RSL3 (p < 0.0001). In PDT (+) LuBi cells, cell survival at 1.25, 2.5, 5, and 10 μM RSL3 was significantly lower than that at 0 μM RSL3 (p < 0.0001, for all). Cell survival at 2.5, 5, and 10 μM RSL3 was significantly lower than that at 1.25 μM RSL3 (p < 0.0001, for all) (Fig. 6B).”

We have also described the cytotoxicity of RSL3 in the discussion.

L298 “treated without PDT” may find a better expression…

Response: Thank you for the suggestion. We have reworded it, as follows:

“treated without PDT”→“in PDT (−)”

L313 to 316 I do not completely agree. Apart from that, tumor cell selectivity of PpIX accumulation is not the crucial point of your investigations – on the contrary!

Response: Thank you for your suggestion. We have removed “5-ALA is a naturally occurring amino acid, the first compound in the heme synthesis pathway [10]. Although exogenous 5-ALA is converted to heme in normal cells, malignant cells preferentially accumulate the active photosensitizer PpIX without being metabolized to produce the heme.”

L327 to 328 this sentence does not make much sense to me

Response: Thank you for the comment. We have revised the sentence, as follows:

“Therefore, it was considered that photobleaching occurred in HDC and LuBi cells during irradiation.”

L348 to 357 this paragraph needs to be completely rewritten. Do have an idea, why RSL3 is so much more toxic to LuBi cells than to HDC cells?

We have rewritten the sentences, as follows:

“Ferroptosis is induced mainly by the small molecules, erastin or RSL3 [25]. RSL3 inhibits GPx4 activity by covalent bonding and leads to lipid peroxide accumulation (Fig 9) [25,26]. It was reported that GPx4 inhibition resulted in ferroptotic death in therapy-resistant cancer cells [27]. The cytotoxicity of RSL3 in HDC cells was increased, at over 5 μM RSL3 (Fig. 6). In addition, the cytotoxicity of RSL3 in LuBi cells was increased, at over 1.25 μM RSL3 (Fig. 6). The difference in cytotoxicity of RSL3 between the two primary tumor cell lines might be correlated with the difference in cellular GPx activity (Fig 5). It was reported that RSL3 enhanced cisplatin treatment for drug-resistant cells, compared with cisplatin treatment alone [28]. The mechanism was that co-treatment with RSL3 and cisplatin remarkably increased malondialdehyde and ROS levels, lipid oxidation, and sensitivity to cisplatin. In this study, although enhanced cytotoxicity due to the combined use of RSL3 and 5-ALA-mediated PDT was observed in HDC cells, this enhanced cytotoxicity was not observed in LuBi cells. Although GPx4 itself might play a role in the cytotoxicity in LuBi cells, it was considered that a mechanism other than that involving GPx4 might be associated with the resistance to 5-ALA-mediated PDT.”

L373 “estimated”?

Response: Thank you for your suggestion. We have reworded it, as follows:

“estimated”→“evaluated”

L358ff I do not properly understand the NO mediated pathways. I just cannot resolve the contradiction of your observation that an NO donor should improve PDT, whereas others claim the contrary and improve PDT effect by blocking NO production (Fahey and Girotti, Cancers 11/2, 2019). May please explain to me – and in the manuscript if necessary.
Response: Thank you for your suggestion.

We have explained it in the discussion section, as follows:

“It has been reported that low levels of endogenous NO promote tumor cell survival, persistence, and progression; they also mediate resistance to chemotherapy and radiotherapy [30,31]. Therefore, it was considered that the resistance of LuBi cells to 5-ALA-mediated PDT might be associated with low NO levels. However, high levels of NO result in cytotoxic effects, thereby promoting apoptosis of tumor cells.”

“It was reported that low levels of NO induced by low PDT light dose were cytoprotective via the upregulation of the anti-apoptotic proteins, NF-κB and Snail, but via the downregulation of Raf kinase inhibitor (RKIP). In contrast, high levels of NO induced by relatively high PDT light doses were cytotoxic via the downregulation of NF-κB and Snail, but the upregulation of RKIP (Fig. 9) [29,30].”

We have also added the following sentences.

“However, we did not evaluate the level of NO, but the percentage of NO positive cells in this study. Therefore, evaluating the relationship between the effect of DETA NONOate and the level of NO should be considered in the future.”

L398 may first state that PpIX levels were similar

Response: Thank you for your suggestion. We have added the sentence, as follows:

“Intracellular PpIX levels before irradiation were similar between the two cell lines.”

L399 between the two cell lines

Response: Thank you for the suggestion. We have rewritten the sentence, as follows:

“…between the two cell lines.”

L401 this is a very week statement

Response: Thank you for your comment. We have added a statement, as follows:

“…under the experimental conditions…”

L405 it is not true that your data show that the NO donor improved PDT efficacy only in the PDT resistant cell line. On the contrary! It worked better in the HDCs.

Response: Thank you for your suggestion. As shown in Fig 7, the percentage of NO-positive cells at pre-PDT was significantly different between the two primary tumor cells. In HDC cells with a higher proportion of NO-positive cells at pre-PDT, NO donor at lower concentration enhanced the cytotoxicity for HDC cells. Contrarily, in LuBi cells with fewer NO-positive cells at pre-PDT, NO donor at higher concentration enhanced the cytotoxicity for treated cells.

We have added the sentence, as follows:

“It was considered that the difference might be correlated with the percentage of NO-positive cells at pre-PDT.”

L415 ??

Response: Thank you for the query. We have removed “The funders had role in the analyses of data.”